# Research of Fluridone’s Effects on Growth and Pigment Accumulation of *Haematococcus pluvialis* Based on Transcriptome Sequencing

**DOI:** 10.3390/ijms23063122

**Published:** 2022-03-14

**Authors:** Jinpeng Sun, Jiawei Zan, Xiaonan Zang

**Affiliations:** Key Laboratory of Marine Genetics and Breeding, Ministry of Education, Ocean University of China, Qingdao 266003, China; sunjinpeng@stu.ouc.edu.cn (J.S.); 13589030343@163.com (J.Z.)

**Keywords:** *Haematococcus pluvialis*, transcriptome sequencing, fluridone, growth, astaxanthin

## Abstract

*Haematococcus pluvialis* has high economic value because of its high astaxanthin-producing ability. The mutation breeding of *Haematococcus pluvialis* is an important method to improve the yield of astaxanthin. Fluoridone, an inhibitor of phytoene dehydrogenase, can be used as a screening reagent for mutation breeding of *Haematococcus pluvialis*. This study describes the effect of fluridone on the biomass, chlorophyll, and astaxanthin content of *Haematococcus pluvialis* at different growth stages. Five fluridone concentrations (0.00 mg/L, 0.25 mg/L, 0.50 mg/L, 1.00 mg/L, and 2.00 mg/L) were set to treat *Haematococcus pluvialis*. It was found that fluridone significantly inhibited the growth and accumulation of astaxanthin in the red dormant stage. In addition, transcriptome sequencing was used to analyze the expression of genes related to four metabolic pathways in photosynthesis, carotenoid synthesis, fatty acid metabolism, and cellular antioxidant in algae after fluridone treatment. The results showed that six genes related to photosynthesis were downregulated. *FPPS*, *lcyB* genes related to carotenoid synthesis are downregulated, but carotenoid β-cyclic hydroxylase gene (*LUT5*), which plays a role in the conversion of carotenoid to abscisic acid (ABA), was upregulated, while the expression of phytoene dehydrogenase gene did not change. Two genes related to cell antioxidant capacity were upregulated. In the fatty acid metabolism pathway, the acetyl-CoA carboxylase gene (*ACACA*) was downregulated in the green stage, but upregulated in the red stage, and the stearoyl-CoA desaturase gene (*SAD*) was upregulated. According to the transcriptome results, fluridone can affect the astaxanthin accumulation and growth of *Haematococcus pluvialis* by regulating the synthesis of carotenoids, chlorophyll, fatty acids, and so on. It is expected to be used as a screening agent for the breeding of *Haematococcus pluvialis*. This research also provides an experimental basis for research on the mechanism of astaxanthin metabolism in *Haematococcus pluvialis*.

## 1. Introduction

*Haematococcus pluvialis* is a kind of unicellular green algae belonging to Chlorophyta, Volvoxales. It has high economic value and research value because it can accumulate astaxanthin in large quantities under stress [1]. Astaxanthin is a type of ketone carotenoid, insoluble in water, but soluble in organic solvents. The conjugated double bond in the astaxanthin molecular structure as well as the unsaturated keto and hydroxyl at the end of the chain can attract unpaired electrons or contribute electrons to free radicals, thus scavenging free radicals in organisms. Its antioxidant capacity is 10 times that of natural β-carotene and 550 times that of natural vitamin E. Therefore, astaxanthin has a very broad application prospect in medical care, cosmetics, feed additives, and other aspects [2,3,4,5]. As the best source of astaxanthin, it is of great value to further explore the potential of astaxanthin production by breeding of *Haematococcus pluvialis*. Mutation breeding is suitable for *Haematococcus pluvialis* because of its single-cell characteristics. Herbicides are often used as screening conditions to improve the screening efficiency of mutants. Substances such as fluridone, diclofenac, quizalofop, compactin, diphenylamine, and nicotine have all been used as screening agents [6,7]. Among them, fluridone (1-methyl-3-phenyl-5-(3-trifluoromethyl) phenyl)-4(1H)-pyridinone), with the molecular formula C_19_H_14_F_3_NO, is a widely used herbicide. Fluridone inhibits plant phytoene dehydrogenase activity, which in turn inhibits the biosynthesis of photoprotective carotenoids, and eventually leads to chlorophyll degradation and plant albino death [8]. The use of fluridone on algae has also been reported. At 25 °C, fluridone can significantly inhibit the photosynthesis of *Chlamydomonas reinhardtii*. Moreover, the content of carotenoids such as β-carotene in the algae treated with fluridone was significantly reduced [9]. On the other hand, studies have shown that non-lethal doses of fluridone could lead to the accumulation of phytoene in *Chlorococcum* sp. The fatty acid composition in *Chlorella* also changed after fluridone treatment [10].

The transcriptome is a collection of RNAs transcribed from a specific tissue or cell at a certain developmental stage or in a certain functional state. By studying the transcriptome, the gene expression level can be understood at the overall level, and the molecular mechanism of biological gene expression regulation can be revealed [11]. There have been many studies on the transcriptome analysis of *Haematococcus pluvialis*. Zhang et al. used transcriptome sequencing to identify differential genes in *Haematococcus pluvialis* at different growth stages, and described the changes in the expression levels of genes such as *LUT1*, *crtR-3*, *lcyB*, *lcyE*, *BKT*, and *CrtR* [12]. Gao et al. used transcriptome sequencing technology to explore the effects of jasmonic acid and salicylic acid on *Haematococcus pluvialis* [13]. Cheng et al. used transcriptome sequencing technology to explore the carbon metabolism pathway of *Haematococcus pluvialis* after mutation [14]. Furthermore, He et al. studied the synergistic effects of high light stress, acetate, and Fe^2+^ on *Haematococcus pluvialis* by transcriptome sequencing, respectively [15]. In this study, transcriptome sequencing technology was used to comprehensively describe the effects of fluridone on *Haematococcus pluvialis* in combination with the appearance morphology, biomass, pigment content, and other indicators. This work lays a theoretical foundation for using fluridone in the mutation breeding of *Haematococcus pluvialis* with a high content of astaxanthin, and also provides the experimental basis for research on the mechanism of astaxanthin metabolism in *Haematococcus pluvialis*.

## 2. Results

### 2.1. Effects of Fluridone on Cell Status and Biomass

Fluridone can significantly inhibit the growth of *Haematococcus pluvialis*, as shown in Figure 1a. In the green stage of *Haematococcus pluvialis*, the inhibitory effect of fluridone was more obvious, and the cell density of algal fluid decreased with the increase in fluridone concentration. After 12 days of induction under high-light and nitrogen-deficient conditions, the control group untreated with fluridone was orange-red due to a large amount of astaxanthin, and most of the cells lost their ability to move and deposited on the bottom of the flask; the experimental group treated with a higher concentration of fluridone was still green, there were still many motile cells in the algae liquid, and the cells were more uniform in color and did not sink. Microscopic observation was carried out on the algal fluid of the control group and the 1.00 mg/L fluridone treatment group, and the cell state of other experimental groups was similar in the microscope field. As shown in Figure 1b, most of the algal cells in the control group were transformed into orange-red cysts that accumulated a large amount of astaxanthin. Although most of the algal cells in the experimental group were also transformed into spherical dormant forms, the cells were still mainly green, and a large number of residues produced by cell lysis could be seen in the field of vision.

### 2.2. Effects of Pigments

The content of chlorophyll a, chlorophyll b, and astaxanthin in *Haematococcus pluvialis* was continuously measured over 12 days of induction. Figure 2a shows the change in chlorophyll content per unit number of cells in the green stage. It can be seen that the content of chlorophyll a and chlorophyll b in the experimental group continued to decrease with time, and was significantly lower than that in the control group. Figure 2b shows the change in chlorophyll in the red stage, as shown in the figure, after entering the induction stage; the content of the two chlorophylls in the unit number of cells increased briefly on the second day of induction, and then continued to decrease. The change trend of the experimental group and the control group was similar, and there was no significant difference. Figure 2c shows the content of intracellular astaxanthin in the unit number of cells treated with different concentrations of fluridone after 12 days of induction. The results showed that the content of intracellular astaxanthin after treatment with fluridone decreased significantly compared with the control group, but the difference between the experimental groups with different concentrations of fluridone was not significant.

### 2.3. Transcription Sequencing Quality and Splicing Information

Through RNA sequencing and data filtering, about 20 million clean reads were obtained for each sample. After assembly by splicing and clustering, 46,785 unigenes longer than 200 bp were obtained. From the unigene length distribution map (Appendix A), it can be concluded that the number of unigenes generally decreases with the increase in length, and there were 8279 genes with a length of more than 2 kbp.

### 2.4. Feature Annotation Results

To investigate the functional information of unigenes (46,785 entries), they were annotated in seven databases, respectively. Overall, 21,704 unigenes (48.88%) were annotated in at least one database, and 1177 unigenes were annotated in all seven databases. The number of unigenes annotated in each database is shown in Table 1. Five databases, GO, KOG, PFAM, NT, and NR, were selected for the Venn diagram results as shown in Figure 3. There were 1385 unigenes annotated in all five databases. Only 2641 entries were annotated in the NR database, 1540 entries were annotated only in the NT database, and no unigenes were annotated only in KOG, GO, and PFAM.

#### 2.4.1. NR Annotated Species Classification

Through the annotation results of the NR database, the species distribution map on the alignment was drawn, as shown in Appendix A. The most aligned was *Chlamydomonas eustigma*, accounting for 30.8%. This was followed by *Chlamydomonas reinhardtii*, *Gonium pectorale*, *Volvox carteri*, and *Raphidocelis subcapitata* in the proportions 20.3%, 11.2%, 9.3%, 5.8%, respectively. Other species accounted for 22.7%.

#### 2.4.2. GO Classification

All 16,653 unigenes successfully annotated into the GO database were classified according to three categories: Biological Process, Cellular Component, and Molecular Function, and 41 subcategories were obtained. In the subcategories of biological processes, the top three were cellular process, metabolic process, and biological regulation. The number of unigenes was 9801 (58.85% of the total), 8504 (51.07%), and 3750 (22.52%), and some Unigenes could be annotated to multiple subclasses. In the subclasses of cellular components, the top three were cellular anatomical entity, intracellular, and protein-containing complex. The number of genes was 9039 (54.28%), 5187 (31.15%), and 3941 (23.67%), respectively. In the molecular function subclass, the top three were binding, catalytic activity, and transporter activity. The number of genes were 7953 (47.76%), 6415 (38.52%), and 1320 (7.93%), respectively. The results are shown in Figure 4.

#### 2.4.3. KOG Classification

As shown in Figure 5, 3866 unigenes were successfully annotated into the KOG database, and all unigenes were classified into 25 categories. The category with the largest number was “post-translational modification, protein turnover, molecular chaperone” with 514 (13.30%), followed by “general function prediction” with 485 (12.55%); the least number of genes were “extracellular”, “structure”, and “cell movement”, and the number of genes was 3 (0.08%) and 6 (0.16%), respectively.

#### 2.4.4. KEGG Classification

A total of 5256 unigenes were successfully annotated in the KEGG Orthology (KO) database. According to the classification method of the KO database, the metabolic pathways involved in genes are divided into five branches, namely: (A) Cellular Processes; (B) Environmental Information Processing; (C) Genetic-Information Processing; (D) Metabolism; and (E) Organic Systems. There were 34 s branches under the first branch. In the first branch, the largest number of genes was classified into “metabolism”-related pathways, followed by “organic systems”. In the second branch, “translation” related pathways had the most genes (465), followed by “signal transduction” (459), and the results are shown in Figure 6.

### 2.5. Analysis of Differentially Expressed Genes

#### 2.5.1. Summary of DEGs under Different Conditions

In order to explore the effect of fluridone treatment on the gene expression of *Haematococcus pluvialis*, two comparison groups were set as the green stage fluoridone-treated group (GF) vs. the green stage control group (GC), and the red stage fluoridone-treated group (RF) vs. the red stage control group (RC), and *p*-adj < 0.05 was used as the standard for differential gene screening. It can be seen from Appendix A that there were 1451 differentially expressed genes in the green stage, of which 435 were upregulated and 1016 were downregulated; there were 3746 differentially expressed genes in the red stage including 1446 upregulated and 2300 downregulated. The number of differential genes in the red stage treated with fluridone was significantly more than those in the green stage. This indicates that in the red stage of the massive synthesis of carotenoids such as astaxanthin, fluoridone as a phytoene dehydrogenase inhibitor has a greater impact on cellular gene expression. The number of downregulated genes in *Haematococcus pluvialis* at different stages was significantly higher than that of the upregulated genes after treatment with fluridone.

As shown in the Venn diagram (Appendix A), there were 259 differential genes co-expressed in the green stage and the red stage after treatment with fluoridone.

By clustering the differential genes of the two comparison groups, a cluster map of gene expression under each experimental condition was obtained (Figure 7). It can be seen from the figure that treated with fluridone, the expression of some genes in the green stage cells and red stage cells were similar, but overall, the difference was significant in gene expression between the two stages of cells. Moreover, there were also significant differences in gene expression between cells in the red stage and cells in the green stage.

#### 2.5.2. GO and KEGG Enrichment Analysis of Green Stage DEGs

In the green cell group (GFvsGC), there were a total of 1451 differential genes including 435 upregulated genes and 1016 downregulated genes. The total differential genes were classified by GO enrichment (corrected *p* value < 0.05), and the results showed that most of the differential genes were significantly enriched in the categories of Cellular Composition (CC) and Biological Process (BP), specifically including “nucleoplasm”, “mRNA processing”, etc.

#### 2.5.3. KEGG Enrichment Analysis of Red Stage DEGs

A total of 3746 differential genes were obtained from the red cell group (RFvsRC), of which 1446 were upregulated and 2300 were downregulated. The red stage KEGG database annotation results showed that the upregulated differential genes were significantly enriched into 15 metabolic pathways, mainly including “arginine and proline metabolism”, “carbon metabolism”, “arginine biosynthesis”, “tricarboxylic acid cycle”, “nitrogen metabolism”, and “amino acid biosynthesis”, etc. The downregulated genes were significantly enriched into five metabolic pathways including “viral oncogenes”. In general, fluridone treatment mainly affected pathways such as amino acid metabolism and carbon metabolism in the red stage of *Haematococcus pluvialis*.

### 2.6. Effects of Fluridone on Specific Metabolic Pathways of Haematococcus pluvialis

Considering the inhibition of astaxanthin synthesis by fluridone and the protective effect of astaxanthin on *Haematococcus pluvialis* cells, based on the KEGG metabolic pathway analysis, a total of four metabolic pathways were selected for analysis including “carotenoid biosynthesis”, “fatty acid metabolism”, “cellular antioxidant capacity”, and “photosynthesis”.

#### 2.6.1. Effects on Carotenoid Biosynthesis

Carotenoid β-cyclic hydroxylase gene (*LUT5*), lycopene β-cyclase gene (*lcyB*), and farnesyl pyrophosphate synthase gene (*FPPS*) of the metabolic pathway of carotenoid synthesis were selected as the differentially expressed genes. No upregulation or downregulation of the phytoene dehydrogenase gene was found in both groups of experiments, indicating that the inhibitory effect of fluridone on phytoene dehydrogenase was not at the transcriptional level. Lycopene β-cyclase (*LcyB*), a key enzyme in the conversion of lycopene to β-carotene, was detected to be downregulated in the RFvsRC group, but no difference was detected in the GFvsGC group, which may be because the synthesis process of carotenoids in the red stage was significantly stronger than that in the green stage, and the inhibitory effect of fluridone was highlighted. In the GFvsGC group and RFvsRC group, the expression of the carotenoid β-cyclic hydroxylase gene (*LUT5*) was upregulated, and carotenoid β-cyclic hydroxylase catalyzed the metabolic pathway of lycopene to lutein. The *LUT5* gene was upregulated when carotenoid synthesis was inhibited. This result may be due to the role of the *LUT5* gene in the abscisic acid (ABA) synthesis pathway. Abscisic acid acts as an anti-stress hormone to help cells resist fluridone stress. The farnesyl pyrophosphate synthase gene catalyzes the condensation of isopentenyl pyrophosphate (IPP) and dimethylallyl pyrophosphate (DAMPP) to form geranyl pyrophosphate (GPP), and then catalyzes GPP to form farnesyl pyrophosphate phosphoric acid (FPP). Farnesyl pyrophosphate is a precursor for carotenoid synthesis. Our experimental results showed that when the downstream carotenoid synthesis pathway was inhibited, the accumulation of intermediate products led to the downregulation of the expression of the farnesyl pyrophosphate synthase gene. It can be speculated that there was also a negative feedback regulation mechanism in this pathway. An overview of the results is shown in Figure 8.

#### 2.6.2. Effects on Fatty Acid Metabolism

Previous research has shown that there is a significant correlation between the accumulation of astaxanthin and oil in *Haematococcus pluvialis*. After treatment with fluridone, the rate-limiting enzyme acetyl-CoA carboxylase gene (*ACACA*) in the de novo fatty acid synthesis pathway was downregulated in the green stage, but upregulated in the red stage (Figure 9a). It is speculated that this difference may be caused by the strong inhibition of the photosynthesis and other related reactions of *Haematococcus pluvialis* by fluridone in the green stage. Stearoyl-CoA desaturase (*SAD*), a rate-limiting enzyme required for the production of monounsaturated fatty acids from saturated fatty acids, was upregulated after fluridone treatment compared to the controls (Figure 9b). It is speculated that fluridone treatment will cause the change in fatty acid content by regulating the activity of key enzymes of fatty acid synthesis. In the red stage, the expression related to fatty acid synthesis is upregulated, which means that the content of fatty acids may increase. Previous studies generally believed that astaxanthin and fatty acid content were related, and that they would change synchronously [16]. However, in this research of fluridone treatment, the content of astaxanthin decreased in the red stage, which is different from the increase of fatty acids. This indicates that the decrease in astaxanthin synthesis under fluridone treatment is mainly due to the direct inhibition of fluridone on the astaxanthin synthesis pathway, rather than through the influence on the fatty acid synthesis pathway.

#### 2.6.3. Effects on Cellular Antioxidant Capacity

As an important antioxidant in cells, carotenoids are of great significance for protecting cells from reactive oxygen species (ROS) damage. Astaxanthin is the carotenoid with the strongest antioxidant capacity at present. The inhibitory effect of fluridone on carotenoids such as astaxanthin is bound to affect the antioxidant capacity of cells. As shown in Figure 10, we analyzed the expression of related genes in the glutathione metabolic pathway, which is also an antioxidant, and found that glutamate cysteine ligase (*GCLC*) in the red stage and serine O-acetyltransferase (*cysE*) in the green stage were upregulated. The results indicate that when the content of carotenoids such as astaxanthin decreased, cells would improve their resistance to reactive oxygen species (ROS) to protect themselves by promoting the synthesis of other antioxidants such as glutathione.

#### 2.6.4. Effects on Photosynthesis

Photosynthesis related genes are greatly affected by fluridone. The experimental results showed that after treatment with fluridone, the genes *Lhcb2* and *Lhcb4* related to photosynthesis were downregulated in the green stage; *chlG*, *Lhca1*, *Lhca2*, and *Lhca4* were downregulated in the red stage (Figure 11). The downregulation of photosynthesis related genes may be inhibited by fluridone or related to the decrease in carotenoid content. Carotenoids, as important light-harvesting pigments, play an important role in the photosynthesis process and are also important photoprotective pigments. The decrease in carotenoid content may cause the change in pigment composition in the photosynthetic system, and then reduce the demand for chlorophyll. This leads to the inhibition of chlorophyll synthesis, which decreases the cell photosynthesis and cell growth, as presented Section 2.1.

### 2.7. Real-Time PCR Verification

For the four metabolic pathways of photosynthesis, carotenoid synthesis, fatty acid metabolism, and cellular anti-oxidation, three genes, *cysE*, *LUT5*, and *ACACA* in the green stage and five genes, *lcyB*, *Lhca2*, *GCLC*, *chlG*, and *ACACA* in the red stage, were selected for analysis. Real-time fluorescence quantitative PCR verification, the results of which are shown in Figure 12, was consistent with the transcriptome sequencing results. The results proved that the transcriptome results were effective. The expression of these genes did change with fluridone treated *Haematococcus pluvialis*, which may be related to its growth and astaxanthin synthesis.

## 3. Discussion

*Haematococcus pluvialis* has important economic value. Improving the culture conditions and selecting algal species to increase astaxanthin production and reduce production costs are of great significance in promoting the development of a *Haematococcus pluvialis* aquaculture industry. Methods used in *Haematococcus pluvialis* breeding mainly include selective breeding, mutation breeding, genetic engineering breeding, and so on [17]. Among them, mutation breeding is widely used because of its simple operation and low cost. Tjahjono et al. used fluridone as a screening pressure to conduct mutagenesis of *Haematococcus pluvialis*, but did not obtain algal strains with high astaxanthin production [7]. Gene mutations caused by conditions such as ultraviolet light or ethyl methane sulfonate (EMS) are random, and due to the complexity of the metabolic process, the resistant algal strains obtained after screening with herbicides such as fluridone may not show the character of high-yield astaxanthin. Therefore, it is necessary to deeply understand the changes in gene expression in *Haematococcus pluvialis* in response to fluridone stress. Transcriptome sequencing techniques to study gene expression levels has been widely used in recent years [11,12,13,14,15]. By knowing the information of differentially expressed genes, we can precisely regulate certain metabolic processes of cells at the gene level. Combined with molecular biology methods such as homologous recombination and gene editing, the efficiency of breeding and the predictability of mutation results may be greatly improved.

It is traditionally believed that fluridone acts on PDS, which inhibits the synthesis of lycopene and then affects the synthesis of carotenoids. However, in this study, the transcription of PDS was not affected by fluridone in both the red cell stage and green cell stage. It can be seen that fluridone does not act on the transcription stage of PDS, but may act on the activity of enzymes. The effect of fluridone on PDS enzyme activity blocked the synthesis of lycopene, and the lack of precursors resulted in the downregulation of the transcription level of the downstream carotenoid synthesis related gene (*lcyB*). At the same time, due to the blocked synthesis of lycopene, the accumulation of upstream substrate phytoene is caused, and the feedback regulation makes *FPPS* and other genes in the upstream synthesis process downregulated. Thus, the whole carotenoid synthesis pathway was inhibited and the astaxanthin level decreased, which was consistent with the effect on the astaxanthin content of *Haematococcus pluvialis* in the culture experiment (Section 2.2). At the same time, we also observed that the expression of the carotenoid β-cyclic hydroxylase gene (*LUT5*) was upregulated, and that the *LUT5* gene plays a role in the metabolic pathway of converting β-carotene to lutein. This also suggests that the decrease in carotenoid content caused by the downregulation of *lcyB*, *FPPS*, and other genes may lead to the upregulation of other pathways, and it is possible to screen algae strains with a high yield of other products such as lutein. Otherwise, under the screening of fluridone, it may be possible to produce a new metabolic pathway for the synthesis of astaxanthin. The algae strains with higher astaxanthin content screened by fluridone may have further increased astaxanthin content.

In addition, most of the astaxanthin in *Haematococcus pluvialis* is esterified with fatty acid and exists in the form of an astaxanthin ester, up to 4% of the dry weight of the alga [18]. Under stress conditions, the accumulation of lipids in *Haematococcus pluvialis* can reach more than 30% of the dry cell weight, and the types of fatty acids are mainly palmitic acid, stearic acid, oleic acid, linoleic acid, and linolenic acid [19]. It has been reported in the literature that the increase in fatty acid content in *Haematococcus pluvialis* cells has a linear relationship with the accumulation of astaxanthin. The main types of fatty acids produced after induction were oleic acid, palmitic acid, and linoleic acid. The oil was mainly triacylglycerol. The esterified astaxanthin is mainly deposited in oil droplets formed by triacylglycerols [16]. However, Mirash Zhekisheva et al. used astaxanthin synthesis inhibitors (norflurazon and diphenylamine) and lipid synthesis inhibitors (sethoxydim) to study the metabolic relationship between *Haematococcus pluvialis* astaxanthin and lipids and found that the accumulation of lipids dominated by triacylglycerol (TAG) was not tightly coupled with the accumulation of astaxanthin, and they believed that a certain amount of lipids was the initiation condition for astaxanthin esterification [20]. Elucidating the metabolic relationship and regulatory mechanism of *Haematococcus pluvialis* astaxanthin synthesis and oil accumulation is of great significance in promoting *Haematococcus pluvialis* astaxanthin production and exploiting the potential of *Haematococcus pluvialis* as a high-quality oil-producing alga. In this study, we found that the expression level of the acetyl-CoA carboxylase gene (*ACACA*), a key enzyme in the de novo synthesis of fatty acids, was upregulated when the intracellular astaxanthin content was greatly reduced after treatment with fluridone at the red stage. It may also verify that the accumulation of astaxanthin in *Haematococcus pluvialis* is not closely related to fatty acid synthesis. In addition, due to the regulatory effect of fluridone on the fatty acid pathway, it may also be used as a regulator or screening agent to regulate fatty acid synthesis in *Haematococcus pluvialis*.

In addition, the expression of carotenoid β-cyclic hydroxylase gene (*LUT5*) was upregulated. In addition to catalyzing the synthesis of lutein, carotenoid β-cyclic hydroxylase plays a role in the metabolic pathway of converting β-carotene to abscisic acid (ABA) [21]. As an anti-stress hormone, abscisic acid plays an important role in promoting cell dormancy and improving cell resistance to the external environment. Therefore, it is speculated that fluridone may promote the production of endogenous abscisic acid in *Haematococcus pluvialis* to resist the stress.

Transcriptome analysis is a very effective method to discover genes of interest, but it is not enough to only understand the changes in gene transcription levels. To fully understand the mechanism of fluridone’s impact on cells, more verification is needed, for example, changes in upstream and downstream metabolites of enzymes corresponding to related genes and changes in gene structure, etc.

## 4. Materials and Methods

### 4.1. Cultivation of Algae

*Haematococcus pluvialis* was obtained from the Laboratory of Algae Biotechnology and Breeding, Ocean University of China, and the algal species number is H2. Before the experiment, the algal was activated twice, and the two-stage culture mode was adopted. In the green growth stage, the improved BBM growth medium in our laboratory was used. The culture system was 95 mL of liquid medium and 5 mL of activated algae in a 250 mL conical flask. The illumination was 800–1000 lux, the temperature was 23 ± 1 °C, and the light period was 12 h:12 h. In the induction stage, a nitrogen-deficient medium was used. The algal liquid in the green stage was centrifuged and the supernatant was discarded. The algal mud was resuspended with 100 mL of induction medium. The fluridone (Ron’s Reagent, Shanghai, China) concentration gradient was set to 0 mg/L, 0.25 mg/L, 0.5 mg/L, 1.0 mg/L, 2.0 mg/L.

### 4.2. Biomass Determination and Pigment Analysis

The biomass of *Haematococcus pluvialis* was characterized by the number of cells per milliliter of algal liquid, and calculated from the absorbance OD_680_. Chlorophyll content was measured using the Arnon method. The determination of astaxanthin content is to measure OD_490_ after extraction with dimethyl sulfoxide (DMSO), the reference formula: Cast (mg/L) = (4.5 × OD_490_ × V1)/V2, where V1 is the volume of the liquid obtained from the extraction, and V2 is the algae used for the extraction liquid volume [22].

### 4.3. Sample Preparation for Transcriptome Sequencing

Green stage: The initial cell density of the algal liquid was adjusted to 1.8 × 10^5^ cells/mL, 90 mL of the algal liquid was taken after 48 h of fluridone treatment (1.00 mg/L), centrifuged at 1500× *g* for 10 min, the supernatant was discarded, and the algal cell pellet obtained was snap-frozen in liquid nitrogen and stored at −80 °C in a refrigerator.

Red stage: After the cell density reached 3.0 × 10^5^ cells/mL, the *Haematococcus pluvialis* medium was changed to the induction medium with a fluridone concentration of 1.0 mg/L, and placed it under 5000 lux light for induction culture. After 48 h, we took 90 mL of algae liquid, centrifuged at 1500× *g* for 10 min, then discarded the supernatant, and the algal cell pellet obtained was snap-frozen in liquid nitrogen and stored at −80 °C in a refrigerator. Four groups were set up in the experiment: the green stage control group (GC), the green stage fluoridone treatment group (GF), the red stage control group (RC), and the red stage fluoridone treatment group (RF), with three biological replicates in each group.

### 4.4. Construction of cDNA Library and Transcriptome Sequencing

Transcriptome sequencing was performed using the Illumina sequencing platform of Novogene Co. Ltd (Beijing, China). After the total RNA of *Haematococcus pluvialis* was obtained, the mRNA was reverse transcribed into cDNA using the NEB general library building method, and the cDNA of about 370–420 bp was screened for library construction. An Agilent 2100 bioanalyzer was used to detect the insert size of the library. After meeting the expectations, the effective concentration of the library was accurately quantified by qRT-PCR (the effective concentration of the library was higher than 2 nM) to ensure the quality of the library. Illumina NovaSeq 6000 sequencing was performed after the library was qualified. In order to ensure the quality and reliability of data analysis, the original data were filtered, which included removing reads with adapters, removing reads whose base information could not be determined, and removing low-quality reads (reads whose bases with Qphred ≤20 accounted for more than 50% of the entire read length). At the same time, Q20, Q30, and GC content were calculated for the clean data. All subsequent analyses were performed based on clean data.

### 4.5. Transcript Splicing and Gene Function Annotation

Clean reads were spliced using Trinity (v2.6.6) software. Corset, based on Trinity splicing, aggregates transcripts into many clusters according to the shared reads between transcripts, and then combines the transcript expression levels between different samples and the H-Cluster algorithm. Transcripts with differences in expression were separated from the original cluster, and new clusters were established. Finally, each cluster was defined as “Gene” [23,24]. The above-obtained unigenes were functionally annotated in seven databases (Nr, Nt, Pfam, KOG/COG, Swiss-prot, KEGG, and GO).

### 4.6. Differentially Expressed Gene Analysis

The FPKM value (number of fragments per kilobase length per million fragments from a gene) was used to determine the expression pattern of differentially expressed genes under different experimental conditions [25]. The DESeq method was used to screen differential genes, and the screening threshold was set to *p*adj < 0.05 (padj is the corrected *p* value). All differentially expressed genes were mapped to each term of the GO database, the number of genes per entry was calculated, and differentially expressed genes that were significantly enriched compared to the whole genome background were found. The threshold for significant difference was corrected *p*-value < 0.05 (corrected *p*-value < 0.05). Using the hypergeometric test, we identified pathways where differentially expressed genes were significantly enriched for all annotated genes. Pathways with FDR ≤ 0.05 were defined as significantly enriched pathways, and were corrected by the software KOBAS (2.0) and the BH method [26,27].

### 4.7. Quantitative PCR Validation of Gene Expression

Based on the transcriptome results, seven differentially expressed genes related to carotenoid metabolism, fatty acid metabolism, photosynthesis, and cellular antioxidant capacity were selected for quantitative validation of gene expression using qRT-PCR. The internal reference gene was designed according to the large subunit of the ribulose 1,5-bisphosphate carboxylase/oxygenase gene (*Rbcl*).

RNA extraction was conducted using the E.Z.N.A.^®^ Plant RNA Kit (OMEGA, Germany), and the integrity of the 28S and 18S bands was checked by agarose gel electrophoresis to verify the RNA quality. The total RNA was reverse transcribed into cDNA, and the nucleic acid concentration was determined by Nanodrop. qPCR was performed using the TB Green Premix Ex Taq II Kit (Tli RNaseH Plus) (Takara, Dalian, China), and each sample was set up in triplicate. The experimental data were analyzed and plotted according to the RQ = 2^−^^△△Ct^. qPCR genes and primers are shown in Table 2.

## 5. Conclusions

Transcriptome sequencing technology was used to comprehensively describe the effects of fluridone on *Haematococcus pluvialis* in combination with appearance morphology, biomass, pigment content, and other indicators. The results showed that fluridone significantly inhibited the growth of *Haematococcus pluvialis* in the green stage and the accumulation of astaxanthin in the red stage. Moreover, genes related to astaxanthin synthesis and photosynthesis were downregulated after treatment with fluridone, genes related to cellular antioxidant activity were upregulated, and the expression patterns of genes related to fatty acid synthesis were diverse. The results have important reference value for the application of fluridone on the breeding of *Haematococcus pluvialis* and also provide the basis for research on the mechanism of astaxanthin metabolism.

## Figures and Tables

**Figure 1 ijms-23-03122-f001:**
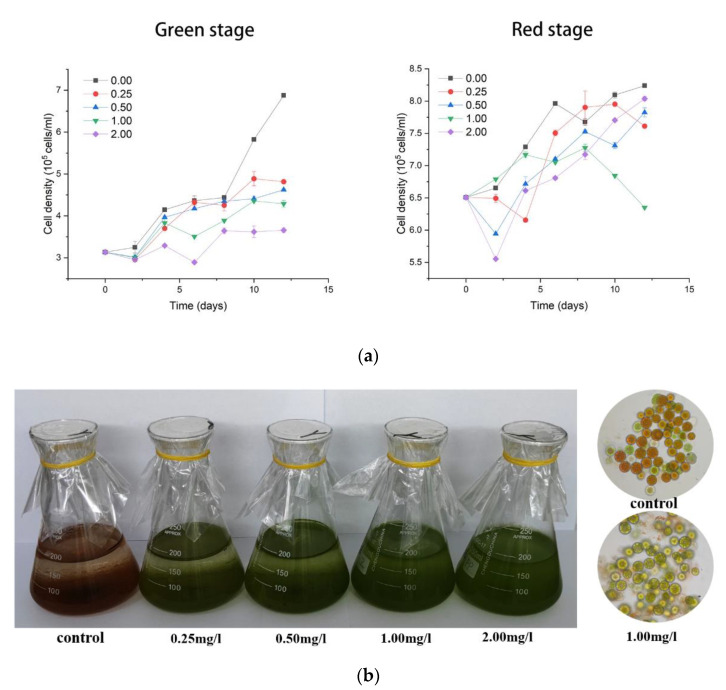
Growth curves and morphological changes of algae treated with different concentrations of fluridone. (**a**) Green stage on the left and red stage on the right. Lines with different colors represent fluridone concentrations of 0.00 mg/L, 0.25 mg/L, 0.50 mg/L, 1.00 mg/L, and 2.00 mg/L. The abscissa represents different days. Values depicted in graphs represent the mean from at least three independent experiments with standard deviations, the same below. (**b**) Observation of algal fluid morphology and microscope field under 0.00 mg/L and 1.00 mg/L treatments.

**Figure 2 ijms-23-03122-f002:**
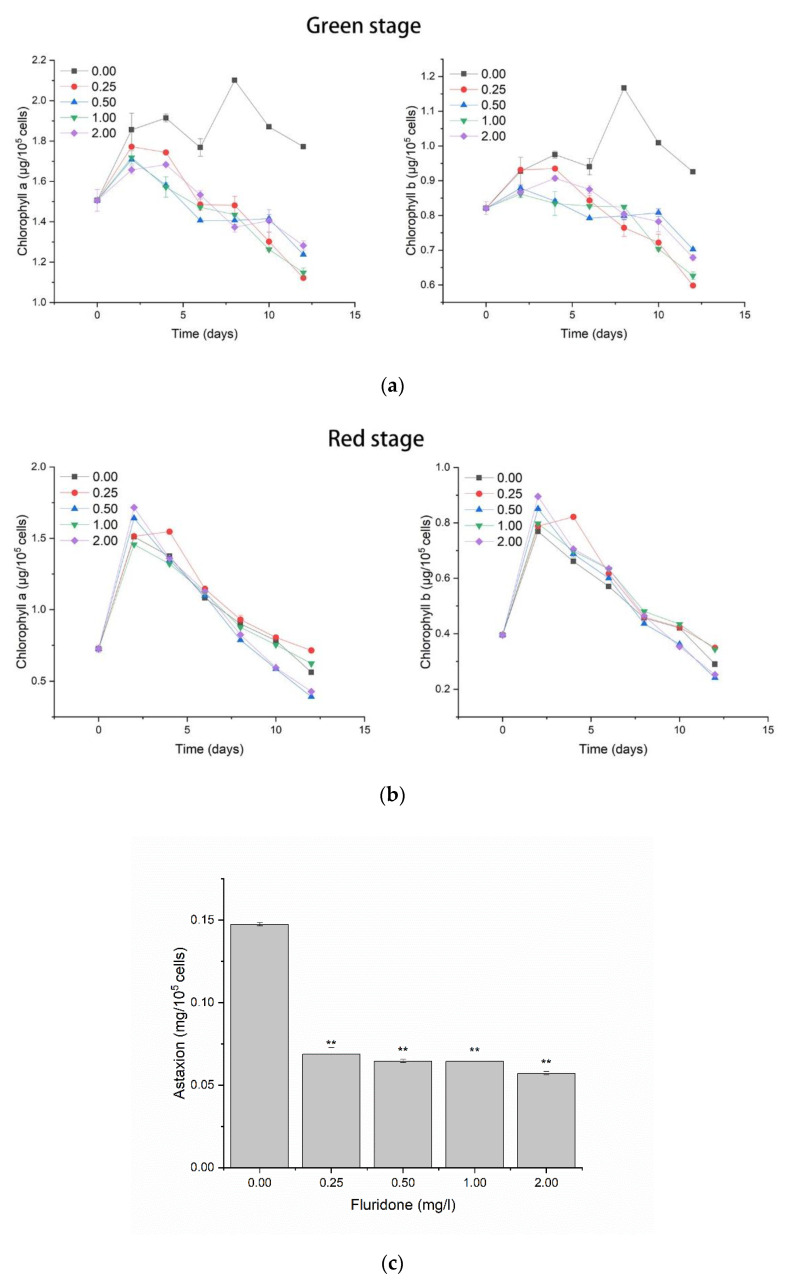
Changes in the content of three pigments after fluridone treatment. (**a**) Changes of chlorophyll a (**left**) and chlorophyll b (**right**) in the green stage, the lines with different colors indicate fluridone concentrations of 0.00mg/L, 0.25mg/L, 0.50mg/L, 1.00mg/L, and 2.00 mg/L. The abscissa represents different days, the same below. (**b**) Changes in chlorophyll a (**left**) and chlorophyll b (**right**) in the red stage. (**c**) Astaxanthin content after 12 days of induction treatment, ** indicates a significant difference from the control group.

**Figure 3 ijms-23-03122-f003:**
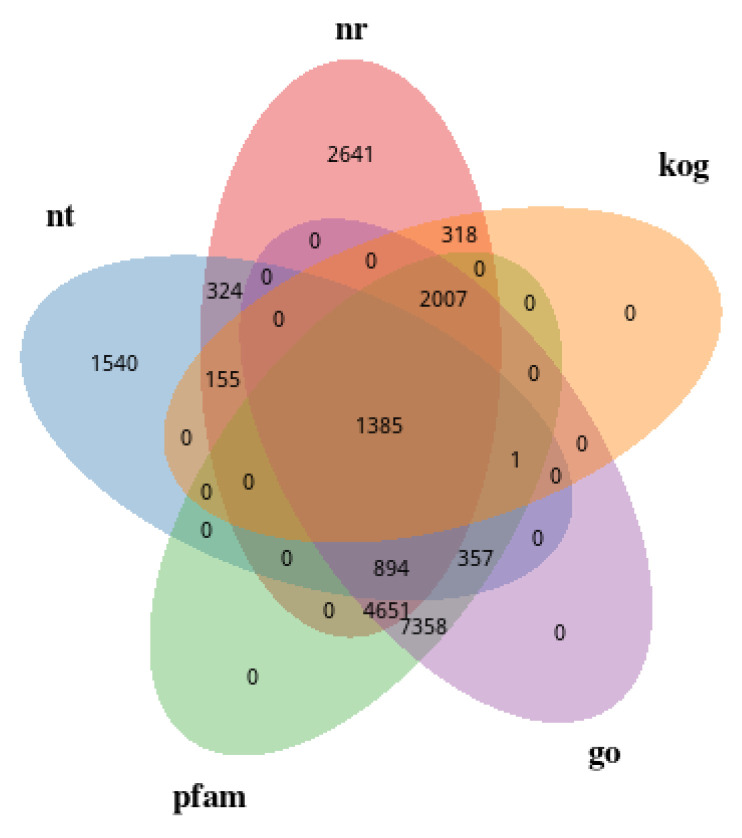
Venn diagram of the annotation results.

**Figure 4 ijms-23-03122-f004:**
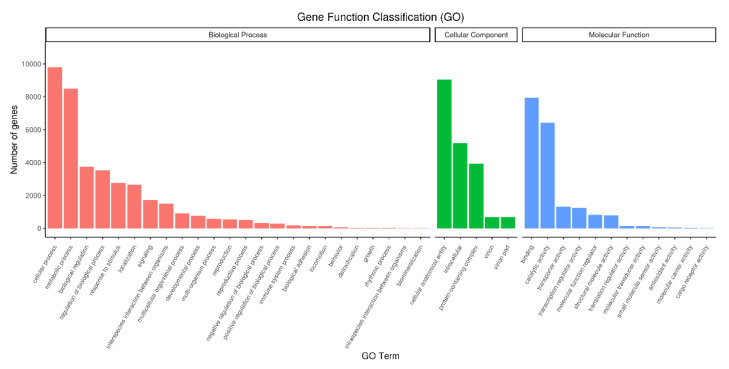
GO Annotation Classification Statistics Chart. The abscissa in the figure is the GO term, and the ordinate is the number of genes annotated to the GO term.

**Figure 5 ijms-23-03122-f005:**
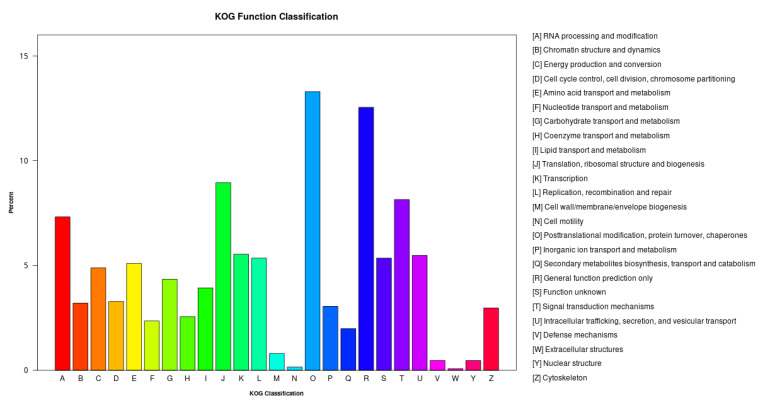
KOG annotation classification chart; the abscissa in the figure is the name of the 26 groups of KOG, and the ordinate is the ratio of the number of genes annotated to this group to the total number of annotated genes.

**Figure 6 ijms-23-03122-f006:**
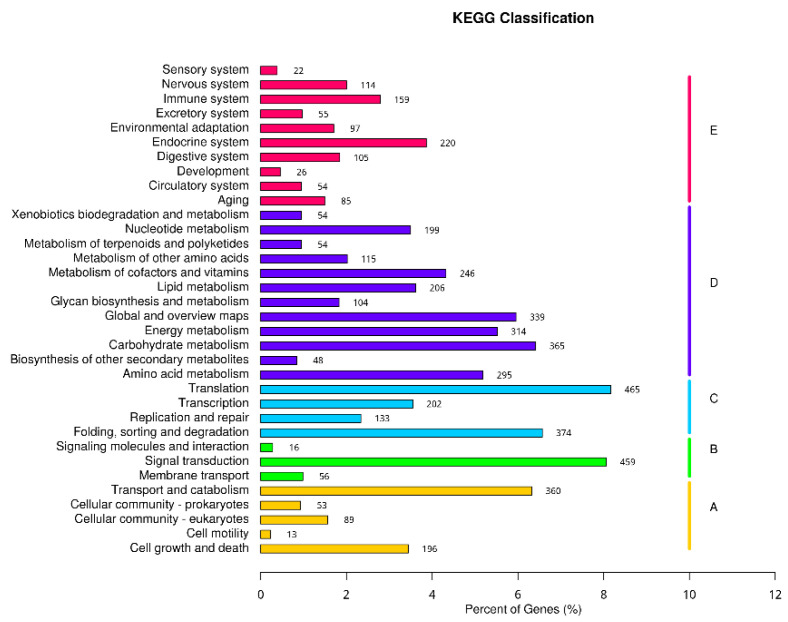
KEGG metabolic pathway classification statistics chart; the ordinate is the name of the KEGG metabolic pathway, and the abscissa is the number of genes annotated under the pathway and the ratio of the number to the total number of genes annotated. The genes are divided into five branches according to the KEGG metabolic pathways involved: Cellular Processes (A, Cellular Processes), Environmental Information Processing (B, Environmental Information Processing), Genetic Information Processing (C, Genetic Information Processing), Metabolism (D, Metabolism), and Organic Systems (E, Organic Systems).

**Figure 7 ijms-23-03122-f007:**
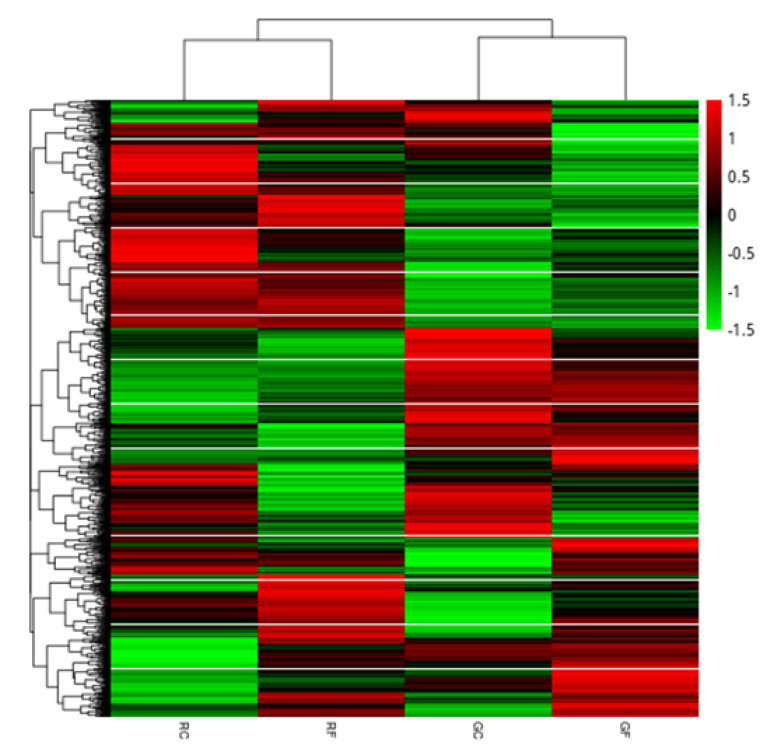
Differentially expressed gene clustering heat map; the abscissa in the figure is the clustering results and the names of samples, the ordinate is the clustering results of differential genes, and the color represents the expression level of the gene in the sample. The redder the color, the higher the expression, and the greener the color, the lower the expression.

**Figure 8 ijms-23-03122-f008:**
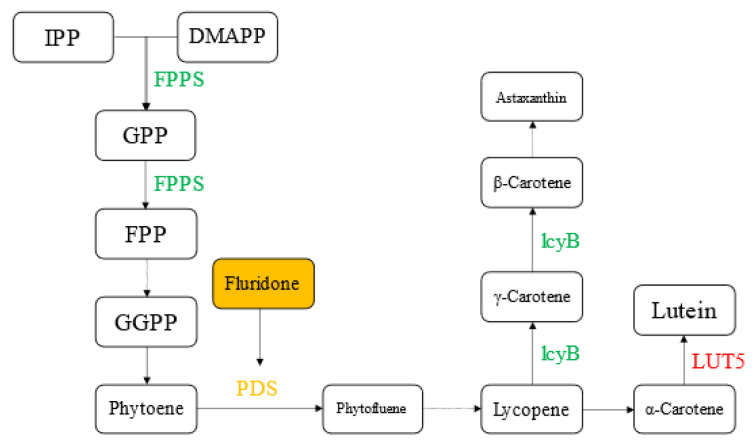
Represents the names of some metabolites and enzymes in the carotenoid metabolic pathway of *Haematococcus pluvialis*. Green indicates downregulation of related enzyme genes, red indicates upregulation, yellow indicates the possible action site of fluridone, and dashed arrows indicate omitted metabolic processes.

**Figure 9 ijms-23-03122-f009:**
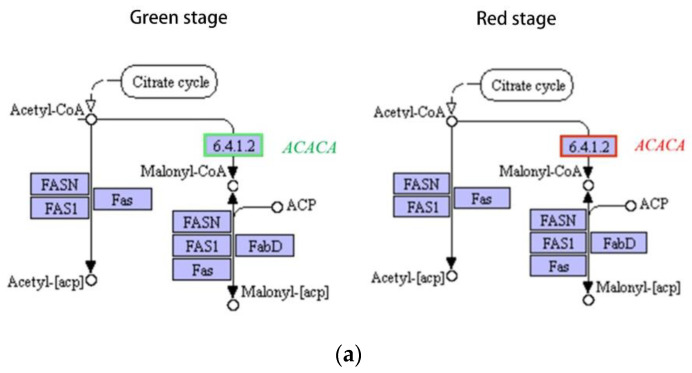
Differences in gene expression of genes related to fatty acid metabolism. Red indicates upregulation and green indicates downregulation. Note: only partial metabolic pathways are shown, the same below. (**a**) Acetyl-CoA carboxylase gene (*ACACA*), green stage on the left and red stage on the right; (**b**) stearoyl-CoA desaturase gene (*SAD*).

**Figure 10 ijms-23-03122-f010:**
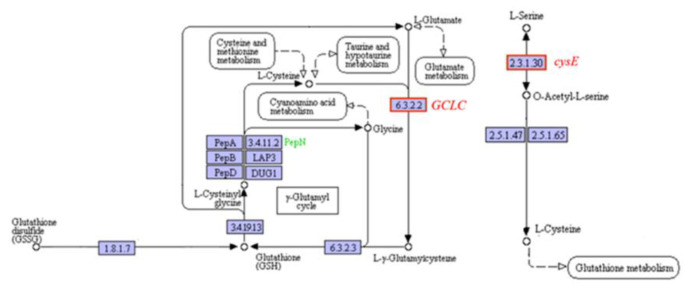
Gene expression of glutamate cysteine ligase gene (*GCLC*) in the red stage and serine O-acetyltransferase gene (*cysE*) in the green stage.

**Figure 11 ijms-23-03122-f011:**
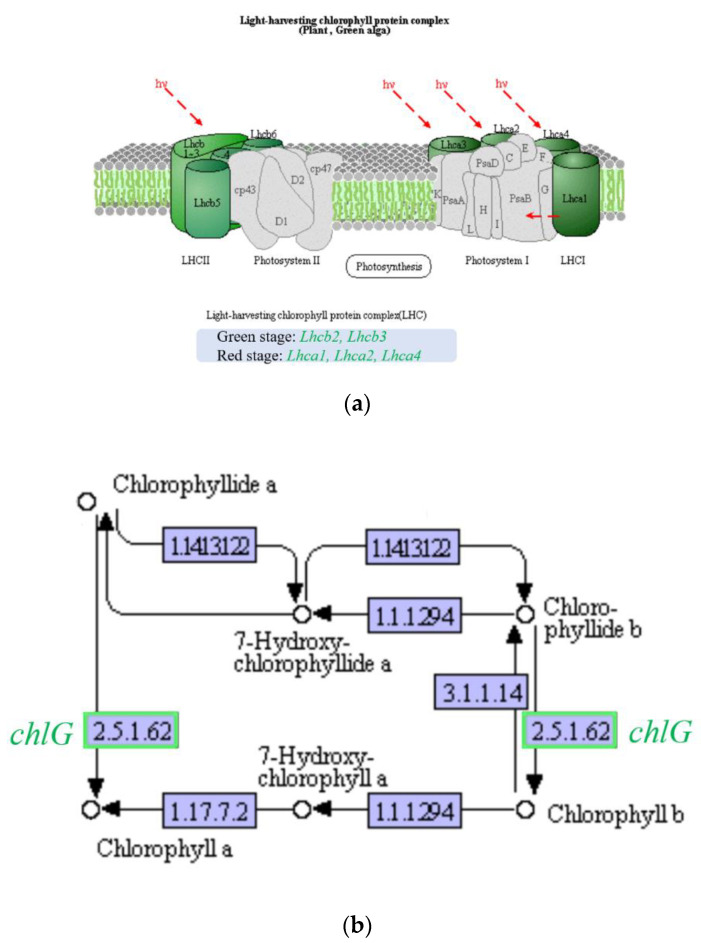
Differentially expressed genes related to photosynthesis: (**a**) Genes related to light harvesting complex; (**b**) Genes related to chlorophyll synthesis.

**Figure 12 ijms-23-03122-f012:**
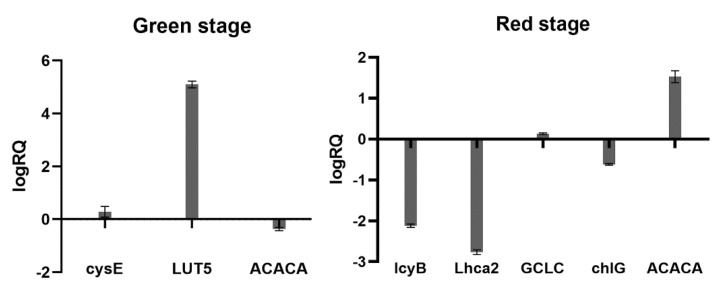
Real-time quantitative PCR verification results, the left side is the gene in the green stage, the right side is the gene in the red stage, the abscissa is the gene name, logRQ > 0 means the gene is upregulated, and logRQ < 0 means the gene is downregulated.

**Table 1 ijms-23-03122-t001:** Gene annotation success rate statistics.

Database	Number of Unigenes	Percentage
Annotated in NR	12,375	26.45
Annotated in NT	4656	9.95
Annotated in KO	5256	11.23
Annotated in SwissProt	7541	16.11
Annotated in PFAM	16,653	35.59
Annotated in GO	16,653	35.59
Annotated in KOG	3866	8.26
Annotated in all Databases	1177	2.51
Annotated in at least one Database	21,704	46.39
Total Unigenes	46,785	100

**Table 2 ijms-23-03122-t002:** List of qPCR genes and primers.

Gene Name	Primers
*Rbcl*	F: ACGAATGTTTACGCGGTGGTCT
	R: GGTACACCCAACTCCTTAGCA
*cysE*	F: TGCCGCCCAGGGACACT
	R: CCCATGCCATCCAGAGCAAG
*LUT5*	F: GATGTCGAGGTCGAGGAGTTGT
	R: CCAGGGATGATGAAGTTAGGG
*ACACA*	F: AAGTAGTAGCAGCCCTCCTCCA
	R: CGCCACCAGAAGATCATAGAA
*lcyB*	F: TGAGCAGGCAACACGAACAGAC
	R: ACCCGCTCTGCTACAGCTATG
*Lhca2*	F: TCGCCCTGCCCAACCAC
	R: TGGTGGGCGCAAAGTACTGGAA
*GCLC*	F: GGGATAATTCCCGCATCAGTTA
	R: GACCTGGTACAAGTGGCTATTC
*chlG*	F: CTCACAGGTTACACCCAGACAA
	R: GATGAAGGAGCCAAACAAGG

## Data Availability

Not applicable.

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
