# Peer review of "Research of Fluridone’s Effects on Growth and Pigment Accumulation of Haematococcus pluvialis Based on Transcriptome Sequencing"

_ijms, 2022, doi:10.3390/ijms23063122_

Round 1

Reviewer 1 Report

I have received the Manuscript entitled: ‘Research of Fluridone’s Effects on Growth and Pigment Accumulation of Haematococcus pluvialis Based on Transcriptome Sequencing (Manuscript ID: ijms-1641893) submitted to the International Journal of Molecular Sciences for a review. The Manuscript describes scientifically significant issue of utilization of transcriptome sequencing technology for description of the effects of fluridone on Haematococcus pluvialis including appearance morphology, biomass, pigment content and other indicators.

The authors correctly justified the choice of Haematococcus pluvialis, which is famus for containing significant amounts of the strong antioxidant known as astaxanthin. Astaxanthin is a widely applied natural nutrient and food additive, due to the fact that there were no harmful effects reported from its consumption. To verify scientific hypothesis authors described the effect of fluridone (applied in various four concentrations from 0.25 mg/L to 2.00 mg/L) on the biomass, chlorophyll and astaxanthin content of Haematococcus pluvialis at different growth stages. In effect it was discovered that that fluridone significantly inhibited the growth of Haematococcus pluvialis in the green stage and the accumulation of astaxanthin in the red stage. Additionally, it was established that genes related to astaxanthin synthesis and photosynthesis were down-regulated after treatment with fluridone. The methodological part is well developed by the authors and the collected results in this field I find scientifically valuable. The discussion in the paper is well-founded and references are well presented. Therefore, I consider the reviewed Manuscript as a good candidate for publication in International Journal of Molecular Sciences. Nonetheless, I recommend to correct three minor errors/ambiguities:

  • Units in whole document should be thoroughly checked and unified, e.g. in lines 440-470 authors use ml with as well as without spacing or there are two different styles for Celsius unit
  • The quality of some graphics is below the acceptance level, therefore, some parts are completely illegible. Particularly, authors should improve the quality of figs. 4-8, 10
  • The article is relatively long, therefore, authors should consider to insert less important data (like some figures) in supporting information, while to most significant will remain in the main article file

Author Response

Point 1: Units in whole document should be thoroughly checked and unified, e.g. in lines 440-470 authors use ml with as well as without spacing or there are two different styles for Celsius unit.

Response 1: Thanks for the suggestions, and we have checked and revised the units in the manuscript. We uniformly added spaces before ml and unified the format of Celsius unit.

Point 2: The quality of some graphics is below the acceptance level, therefore, some parts are completely illegible. Particularly, authors should improve the quality of figs. 4-8, 10.

Response 2: We really appreciate your advice, we have improved the quality of the graphics in the manuscript.

Point 3: The article is relatively long, therefore, authors should consider to insert less important data (like some figures) in supporting information, while to most significant will remain in the main article file.

Response 3: Thanks for your advice, we have put figure 3, figure 5, figure 9 and table 2 into the supplementary materials, and made corresponding modifications in the manuscript.

Reviewer 2 Report

The article is well written and the data and research topic are interesting. The study is useful and could help expand the application of Haematococcus in the future.

However, some minor corrections are requested in order to improve the quality of the manuscript

  • Lines 44-45 Avoid the repletion. Mutagenesis breeding is a very suitable breeding method for Haematococcus pluvialis because it is unicellular organism
  • Line 69 change and with Moreover, Furthermore, or similar
  • Line 89 justify why you put only the picture of sample treated with 1 mg/L of Fluoride. Were the others treated samples like the example reported?
  • In Fig. 1a, 2a, 2b, and 12a put title on the top of the different graph such as “green stage” and “red stage” like you have done in Fig. 15.
  • Line 127 In the caption of fig. 2 A and B in Chlorophyll A and Chlorophyll B should be in lowercase.
  • Line 130 remove very.
  • Lines 132- 205 and Fig. 3-8 should be summarized and put in Materials and methods.
  • Figure 9 and 10 could be put together

Author Response

Point 1: Lines 44-45 Avoid the repletion. Mutagenesis breeding is a very suitable breeding method for Haematococcus pluvialis because it is unicellular organism. 

Response 1: We really appreciate your advice, in order to make the expression more concise, we have modified the sentence to " Mutation breeding is suitable for Haematococcus pluvialis because of its single-cell characteristics."

Point 2: Line 69 change and with Moreover, Furthermore, or similar.

Response 2: Thanks for the suggestions, and we have replaced “And” with “Furthermore”.

Point 3: Line 89 justify why you put only the picture of sample treated with 1 mg/L of Fluoride. Were the others treated samples like the example reported?Response 3: Thanks for the suggestions, we have added an explanation to the manuscript: “Microscopic observation was carried out on the algal fluid of the control group and the 1.00 mg/l fluridone treatment group, and the cell state of other experimental groups was similar in the microscope field.”

Point 4: In Fig. 1a, 2a, 2b, and 12a put title on the top of the different graph such as “green stage” and “red stage” like you have done in Fig. 15.
Response 4:Thanks for your advice, we have added titles to the corresponding figures to distinguish the cell growth stage.

Point 5: Line 127 In the caption of fig. 2 A and B in Chlorophyll A and Chlorophyll B should be in lowercase.

Response 5: We have made corresponding changes. Thanks!

Point 6: Line 130 remove very.

Response 6: We have removed “very”, thank you.

Point 7: Lines 132- 205 and Fig. 3-8 should be summarized and put in Materials and methods.

Response 7: Thanks for the suggestions, lines 132-205 and figure 3-8 are the results of transcriptome sequencing. The methods of transcriptome sequencing have been described in Materials and methods.

Point 8: Figure 9 and 10 could be put together

Response 8: Thanks for the suggestions, in order to make the manuscript more concise, we have put Figure 9 into the supplementary materials in combination with other reviewers' comments.